# The Roles of Mitochondria in Human Being’s Life and Aging

**DOI:** 10.3390/biom14101317

**Published:** 2024-10-17

**Authors:** Hiroko P. Indo, Moragot Chatatikun, Ikuo Nakanishi, Ken-ichiro Matsumoto, Motoki Imai, Fumitaka Kawakami, Makoto Kubo, Hiroshi Abe, Hiroshi Ichikawa, Yoshikazu Yonei, Hisashi J. Beppu, Yukiko Minamiyama, Takuro Kanekura, Takafumi Ichikawa, Atthaphong Phongphithakchai, Lunla Udomwech, Suriyan Sukati, Nurdina Charong, Voravuth Somsak, Jitbanjong Tangpong, Sachiyo Nomura, Hideyuki J. Majima

**Affiliations:** 1Department of Oncology, Graduate School of Medical and Dental Sciences, Kagoshima University, Kagoshima City 890-8544, Japan; hindoh@dent.kagoshima-u.ac.jp (H.P.I.);; 2Amanogawa Galactic Astronomy Research Center (AGARC), Kagoshima University Graduate School of Sciences and Engineering, 1-21-40 Korimoto, Kagoshima 890-0065, Japan; 3School of Allied Health Sciences, Walailak University, Thasala 80161, Thailand; moragot.ch@wu.ac.th (M.C.); suriyan.su@wu.ac.th (S.S.); nurdina.ch@wu.ac.th (N.C.); voravuth.so@wu.ac.th (V.S.); rjitbanj@wu.ac.th (J.T.); 4Research Excellence Center for Innovation and Health Products (RECIHP), School of Allied Health Sciences, Walailak University, Thasala Nakhon Si Thammarat 80160, Thailand; 5Quantum RedOx Chemistry Team, Quantum Life Spin Group, Institute for Quantum Life Science (iQLS), National Institutes for Quantum Science and Technology (QST), 4-9-1 Anagawa, Inage-ku, Chiba 263-8555, Japan; nakanishi.ikuo@qst.go.jp; 6Quantitative RedOx Sensing Group, Department of Radiation Regulatory Science Research, Institute for Radiological Science (NIRS), National Institutes for Quantum Science and Technology (QST), 4-9-1 Anagawa, Inage-ku, Chiba 263-8555, Japan; 7Department of Molecular Diagnostics, School of Allied Health Sciences, Kitasato University, 1-15-1 Kitasato, Sagamihara 252-0373, Japan; 8Department of Applied Tumor Pathology, Graduate School of Medical Sciences, Kitasato University, Sagamihara 252-0374, Japan; 9Regenerative Medicine and Cell Design Research Facility, School of Allied Health Sciences, Kitasato University, 1-15-1 Kitasato, Sagamihara 252-0373, Japan; 10Department of Regulation Biochemistry, Graduate School of Medical Sciences, Kitasato University, 1-15-1 Kitasato, Sagamihara 252-0373, Japan; 11Department of Health Administration, School of Allied Health Sciences, Kitasato University, 1-15-1 Kitasato, Sagamihara 252-0373, Japan; 12Department of Environmental Microbiology, Graduate School of Medical Sciences, Kitasato University, 1-15-1 Kitasato, Sagamihara 252-0373, Japan; 13Department of Medical Life Systems, Graduate School of Life and Medical Sciences, Doshishia University, Kyoto 610-0394, Japan; 14Anti-Aging Medical Research Center and Glycation Stress Research Center, Graduate School of Life and Medical Sciences, Doshisha University, Kyoto 610-0394, Japan; 15Dr. Beppu’s Oral Health Care & Anti-Aging Clinic, Chuo-ku, Tokyo 103-0027, Japan; 16Food Hygiene and Environmental Health Division of Applied Life Science, Graduate School of Life and Environmental Sciences, Kyoto Prefectural University, Sakyo-ku, Kyoto 606-8522, Japan; 17Department of Dermatology, Kagoshima University Graduate School of Medical and Dental Sciences, Kagoshima 890-8544, Japan; 18Division of Nephrology, Department of Internal Medicine, Faculty of Medicine, Prince of Songkla University, Songkhla 90110, Thailand; 19School of Medicine, Walailak University, Thasala 80161, Thailand; 20Hematology and Transfusion Science Research Center (HTSRC), School of Allied Health Sciences, Walailak University, Nakhon Si Thammarat 80160, Thailand; 21Department of Clinical Pharmaceutical Sciences, School of Pharmacy and Pharmaceutical Sciences, Hoshi University, 2-4-41 Ebara, Shinagawa-ku, Tokyo 142-8501, Japan; sachiyo.nomura1012@gmail.com; 22Isotope Science Center, The University of Tokyo, 2-22-16 Yayoi, Bunkyo-ku, Tokyo 113-0032, Japan; 23Department of Gastrointestinal Surgery, The University of Tokyo Hospital, 7-3-1 Hongo, Bunkyo-ku, Tokyo 113-8655, Japan

**Keywords:** mitochondria, aging, mitochondrial DNA, mitochondrial DNA damage, heteroplasmy

## Abstract

The universe began 13.8 billion years ago, and Earth was born 4.6 billion years ago. Early traces of life were found as soon as 4.1 billion years ago; then, ~200,000 years ago, the human being was born. The evolution of life on earth was to become individual rather than cellular life. The birth of mitochondria made this possible to be the individual life. Since then, individuals have had a limited time of life. It was 1.4 billion years ago that a bacterial cell began living inside an archaeal host cell, a form of endosymbiosis that is the development of eukaryotic cells, which contain a nucleus and other membrane-bound compartments. The bacterium started to provide its host cell with additional energy, and the interaction eventually resulted in a eukaryotic cell, with both archaeal (the host cell) and bacterial (mitochondrial) origins still having genomes. The cells survived high concentrations of oxygen producing more energy inside the cell. Further, the roles of mitochondria in human being’s life and aging will be discussed.

## 1. Oxidative Stress, Birth of Earth, and Birth of Living Organisms on Earth

### 1.1. Oxidative Stress

Helmut Sies originated the concept of “Oxidative Stress”; the first publication relating to “Oxidative Stress” appeared in 1982 [1]. Oxidative stress comes from the existence of oxygen, and the influence of oxygen on human beings. We proposed a mitochondrial superoxide theory for oxidative stress diseases and aging in 2015 [2]. A total of 2–3% of electrons of the electron transport chain (ETC) in mitochondria leak, and oxygen binds to the electrons and makes superoxide inside mitochondria (Figure 1). Superoxide then produces many other reactive oxygen species (ROS) and causes apoptosis [2], goes out of mitochondria, and induces intracellular signals in the cytosol (Figure 2) [1,3,4].

The ETC is composed of over 90 subunits. Among the subunits, yellow is the one translated by the genes in mitochondrial DNA. NADH and FADH_2_ carry protons (H^+^) and electrons (e^−^) to the electron transport chain located in the inner membrane. The energy from the transfer of electrons along the chain transports the protons across the membrane and creates an electrochemical gradient. As the accumulating protons follow the electrochemical gradient back across the membrane through an ATP synthase complex, their movement provides energy for synthesizing ATP from ADP and phosphate. At the end of the electron transport chain, two protons, two electrons, and half of an oxygen molecule combine to form water. Then 2–3% of electrons leak from ETC, and oxygen traps the electrons and becomes superoxide (O_2_**^•^**^−^). O_2_**^•^**^−^ can change its form to become various reactive oxygen species (ROS) in mitochondria (Figure transferred from Reference [3]).

Early life was born on Earth and has evolved, and the birth of multicellular lives has occurred. Finally, human beings emerged. Multicellular lives came about because large energy production could begin in cells due to the emergence of mitochondria. A mitochondrion is another cell that enters the other cells and starts symbiosis.

The body develops diseases due to toxic substances associated with the organism. Some toxic substances are linked to the ROS production system and ultimately cause intracellular damage. The body has a defense system against ROS [2,5]. In recent years, the biological defense system, especially the anti-reactive oxygen species (ROS) enzyme system, has been the focus of much attention.

Numerous reports have linked ROS as a factor in many diseases, such as rheumatism, hepatitis, intestinal inflammation, carcinogenesis, and aging [2,5]. Many neurological diseases have also been found to be ROS-related [2,5]. For example, Alzheimer’s, Parkinson’s, amyotrophic lateral sclerosis (ALS), and other neurological diseases are now being recognized as neurological disorders resulting from abnormalities in the body’s defense systems or from increased intracellular levels of ROS. When ROS stress is suffered to the body, it causes damage to cells, which are the smallest units of the body, and this is the cause of the onset of certain diseases. Recently, with the advent of nitric oxide (NO), the reactive nitrogen cascade may also be included in the ROS cascade.

Conversely, the biological defense system has enzymatic and non-enzymatic systems. [2,5]. Thus, the ROS system is detoxified by its scavenging enzyme group. These anti-ROS systems work in unison to combat ROS and aging.

### 1.2. Birth of Earth

#### 1.2.1. The Composition of Earth’s Primitive Atmosphere

The universe began 13.8 billion years ago, and Earth was born 4.6 billion years ago (Figure 3) [6]. Initially, the atmosphere contained little or no free oxygen, and oxygen concentrations increased nearly 2.0 billion years ago [7]. As Earth was born, Earth’s surface rose due to volcanic activity. The composition of Earth’s primitive atmosphere has been investigated for many years because of its possible influence on the origin of life [7]. Kasting used the one-dimensional photochemical model to examine the effect of bolide impacts on the oxidation state of Earth’s primitive atmosphere [7,8]. The impact rate should have been high before 3.8 Ga before the present. Impacts of comets or carbonaceous asteroids should have enhanced the atmospheric CO/CO_2_ ratio by bringing CO ice and/or organic carbon that can be oxidized to CO in the impact plume. Ordinary chondritic impactors would contain elemental iron that could have reacted with ambient CO_2_ to give CO. Nitric oxide (NO) should also have been produced by reaction between ambient CO_2_ and N_2_ in the hot impact plumes. High NO concentrations increase the atmospheric CO/CO_2_ ratio by increasing the rainout rate of oxidized gases [8]. A high atmospheric CO/CO_2_ ratio may have helped facilitate prebiotic synthesis by enhancing hydrogen cyanide and formaldehyde production rates. Formaldehyde may have been produced more efficiently by photochemical reduction of bicarbonate and formate in Fe ^++^-rich surface waters [8].

Magmatic outgassing of volatiles from Earth’s interior probably played a critical role in determining the composition of the earliest atmosphere, more than 4000 Myr ago [9]. Given an elemental inventory of hydrogen, carbon, nitrogen, oxygen, and sulfur, the identity of molecular species in gaseous volcanic emanations depends critically on the pressure of oxygen. Samples from this earliest period of Earth’s history are limited to igneous detrital zircons that pre-date the known rock record, with ages approaching ~4400 Myr ago [9]. Wen et al. examined photochemical reactions in the early Earth’s atmosphere in the prebiotic synthesis of simple organic molecules. They demonstrated that CO_2_ and H_2_ can be converted to simple organic molecules [10].

#### 1.2.2. Reconstructing Ancient Oxygen Levels–Earth and Lives, Emergence of Oxygen

Kump described the prevailing view of atmospheric oxygen evolution over time (Figure 4) [11]. The label oxygen in the air was detected as the degree of mass-independent fractionation (MIF) of sulfur isotopes in rocks. The oxygen was not detectable label up to 2.45 billion years; oxygen levels showed 0.001% of the present atmospheric level (PAL). Rocks older than about 2.45 billion years contain a large MIF of sulfur isotopes; rocks older than 2.45 billion years show essentially none. The accumulation of oxygen and sulfate began in the early Proterozoic eon, 2.5 to 0.54 billion years ago [12]. Cyanobacteria and microscopic algae supply oxygen to the atmosphere. Microorganisms may also have played a major role in atmosphere evolution before the rise of oxygen [13]. Lyons et al. suggested a far more dynamic sequence of events, with the possibility of a much larger increase early on, dating from roughly 2.3 to 2.1 Gyr ago, and then a deep plunge to lower levels that extended over a few hundred million years after the onset of the great oxidation event (GOE) [14,15]. The increased oxygen formed free oxygen (O_2_) and spawned early biological production. The initial increase of O_2_ in the atmosphere, its delayed build-up in the ocean increases to near-modern levels in the sea and air, and its cause-and-effect relationship with life is among the most compelling stories of Earth’s history [14]. Oxygen has toxicity, and the increased amount of oxygen made cells construct antioxidant systems, such as antioxidant enzymes inside their living organism to fight against the oxidants [2].

Rubey (1951, 1955) suggested gases are dominated by CO_2_ rather than CH_4_ [8]. A CO_2_-rich atmosphere is also what one would expect from looking at the atmosphere of Mars, which is 96% CO_2_ [16]. The atmosphere of Venus is composed of 96.5% carbon dioxide [17]. Miller has shown that primitive Earth conditions can synthesize amino acids [18]. Recently, the Hayabusa 2 analyzed the Steroid Ryugu soil sample and found amino acids in the samples [19]. They showed that the asteroid Ryugu is (as expected) rich in carbon, organic molecules, and volatiles (like water) and hinted at the possibility that it was once a comet. A recent analysis shows that Ryugu carries strains of over 15 amino acids [19]. These findings could provide new insight into how life is distributed throughout the cosmos and could mean that it is more common than previously thought.

### 1.3. Birth of Living Organisms on Earth

Approximately 3.5 billion years ago, cyanobacteria had begun to proliferate on the surface, protected from deadly UV rays by greenhouse gasses [20] Cyanobacteria are the only known prokaryotes capable of oxygenic photosynthesis [21]. By 3.46 billion years ago, these photosynthesizing microbes had released significant amounts of oxygen into the atmosphere and oceans [22]. They performed the same functions deep beneath the sea, congregating near undersea volcanoes and thermal vents and reducing metals, minerals, and carbon dioxide [20].

There is ongoing debate in the academic community on the birth of living organisms on Earth. The oldest record of organic matter with a carbon isotope composition consistent with life processes has been reported in 4.1 Ga zircons [23,24], 3.95-Ga metamorphic rocks from Canada [23,25], and the 3.8–3.7 Ga Akilia belt and Isua belt [23,26,27,28,29] in Greenland. Schoph has described the age of the fossiliferous Apex chert as about 3.47 Ga, found in Western Australia [30].

## 2. Mitochondria and Human Being’ Life

### 2.1. Emergence of Mitochondria

#### ATP Production from Mitochondria

The emergence of mitochondria and the birth of reptiles, animals, plants, and eukaryotic one-celled living organisms distinct from multicellular plants and animals: protozoa, slime molds, and eukaryotic alga prokaryotic cells and eukaryotic cells. Mitochondria were the other living organisms that intruded on another living organism and started fateful endosymbiosis more than 1.45 billion years ago [31,32]. The living organisms produced adenosine triphosphate (ATP) using glycolysis. Approximately 2.45 billion years ago, the amount of oxygen started to increase to 1% levels [11]. Mitochondria emerged 1.45 billion years ago, and oxygen levels increased ~1 billion years ago. Then, mitochondria started to make ATP with oxygen and produced over 15 times more ATPs burning oxygen, which is oxidative phosphorylation [2].

Mitochondria have been considered mediators of cell metabolism involved in important life processes, such as aging, cell death, and persistent chronic diseases, in addition to their central function of producing ATP. Chronic diseases cause mutations or deletions in the mitochondrial genome (mt genome), which is believed to accumulate damage from long-term oxidative stress. Mitochondrial DNA (mtDNA) encodes 13 genes for proteins that comprise a part of the electron transport chain (ETC) [5]. Treatment with ETC inhibitors (rotenone, 3-nitro propionic acid, thenoyltrifluoroacetone, antimycin A, and sodium cyanide) generates increased ROS [33]. A significant increase in ROS was also observed in cells lacking mtDNA and mtDNA-deleted cells compared with their parental cells, although no increase was observed in cybrids. Furthermore, cells transfected with cDNA encoding manganese superoxide dismutase had decreased levels of ROS. These results suggest that more ROS are generated from mitochondria in cells with the ETC impaired either by inhibition or damage to the mt DNA.

It is known that combustion requires cleavage of the H-H bond in addition to cleavage of the O-O bond to produce H_2_O. A reaction of hydrogen and oxygen into water vapor releases 242 kJ/mol of heat and reduces the enthalpy accordingly at constant temperature and pressure [34]. Conversely, in mitochondria, the oxidative phosphorylation process (2H^+^ + O_2_ + 2NADH) yields 473 kJ/mol [35]. Oxidation of glucose (C_6_H_12_O_6_ + 6O_2_ → 6CO_2_ + 6H_2_O [36]) produces 2840 kJ/mol from glucose and 6O_2_. Oxidative phosphorylation produces 473 kJ/mol from glucose and O_2_. Glycolysis produces 2 ATP and 96 kJ/mol from 1 glucose [37]. Oxidative phosphorylation produces 38 ATP from glucose [38] versus 2 ATP molecules from glycolysis [36], approximately 5% of glucose’s energy potential. Oxidative phosphorylation produces much more efficient energy in cells.

### 2.2. Ancient Mitochondria Produce Hydrogen

Mitochondria produce hydrogen under anaerobic conditions, in addition to ATP production under aerobic conditions [39,40,41]. The superoxide dismutase was the enzyme that catalyzed H_2_S when the oxygen concentration was much lower than the present time. Reactive sulfur species (RSS) such as H_2_S, HS•, H_2_Sn, (n = 2–7) and H_2_S•^−^ are chemically similar to H_2_O and ROS; HO•, H_2_O_2_, O_2_•^−^ and act on common biological effectors. RSS was kept in evolution long before ROS, and both are metabolized by catalase. It has been suggested that “antioxidant” enzymes evolved to regulate RSS [39]. Although ATP synthesis in mitochondria usually involves the oxidation of reduced carbon compounds, many alpha-proteobacteria and some mitochondria use sulfide (H_2_S) as an electron donor for the respiratory chain and its associated ATP synthesis. In many eubacteria, sulfide oxidation involves the enzyme sulfide: quinone oxidoreductase (SQR). Nuclear-encoded homologs of SQR are found in several eukaryotic genomes.

The eukaryotic SQR was an acquisition from the mitochondrial endosymbiont; from the ancestor of mitochondria [40]. The geological record also contains evidence of life’s history. The permanent oxygenation of the atmosphere is now thought to have occurred 2.2 billion years ago, and large parts of the deep ocean remained anoxic until less than 0.5 billion years ago [42,43]. Eukaryotes existed for at least 1.5 billion years and must have spent much of their evolutionary history in oxygen-poor and sulfide-rich environments [44]. The endosymbiotic origin of mitochondria during eukaryogenesis has long been viewed as an adaptive response to the oxygenation of Earth’s surface environment, presuming a fundamentally aerobic lifestyle for the free-living bacterial ancestors of mitochondria. This oxygen-centric view has been robustly challenged by recent advances in the Earth and life sciences [45]. Eukaryotes arose about 1.6 billion years ago, when oxygen levels were still very low on Earth, both in the atmosphere and the ocean [45]. It is, therefore, natural that many lineages of eukaryotes harbor and use enzymes for oxygen-independent energy metabolism. The absence of oxygen offers energetic benefits of the same magnitude as the presence of oxygen. Superoxide dismutase—localized in mitochondria and the cytosol, detoxicate superoxide, and byproducts from the mitochondrial electron transport chain (ETC)—was the machinery that catalyzed H_2_S as its substrate [40]. Some eukaryotes evolved to live in permanently oxic environments (life on land).

In contrast, other eukaryotes have remained specialized to low oxygen habitats, where the Km of mitochondrial cytochrome c oxidase is 0.1–10 μM for O_2_, which corresponds to about (average 0.4%) of present atmospheric O_2_ levels [45]. Hjort et al. found important insights into eukaryote molecular cell biology and evolution by studying eukaryotic diversity from the perspective of their mitochondrial variants. These investigations contribute to understanding the essential functions of mitochondria, defined in the broadest sense, and the limits to which reductive evolution can proceed while maintaining a viable organelle [46]. The development and evolution of mitochondria have taken a long time to mature or differ among species. Javaux may indicate that mitochondria join the parent cells after the first Eukaryote common ancestor (FECA) was developed [23].

After oxygen arose in the atmosphere, living things started to face the danger of oxygen, a harmful substance that oxidized things. Then, cells obtain the antioxidant machinery to protect themselves against oxygen. The cells were efficient and evolved to use oxygen to produce more energy inside the body; that was the evolved machinery, mitochondria. Mitochondria, as eukaryotic origins, lie within the Archaea domain and α-Proteobacteria [32,47]. Studying eukaryotic diversity from the perspective of their mitochondrial variants has yielded important insights into eukaryote molecular cell biology and evolution [46].

### 2.3. Lifespan in Different Species

Perez-Campo et al. (1998) examined the relationship of oxidative stress with maximum life span (MLSP) in different vertebrate species: pig-tailed macaque, rhesus monkey, baboon, chimpanzee, hamster, rat, sheep, pig, Guinea pig, frog, trout, Xenopus, canary, pigeon, and human beings [48]. They showed that the endogenous levels of catalase, glutathione (GSH) peroxidase, GSH reductase, and hydrogen peroxide products in tissues negatively correlate with MLSP [48]. The most longevous animals studied in each group, pigeon or man, show the minimum levels of mitochondrial oxygen radical production. They indicated that the low rate of mitochondrial oxygen radical production could decrease oxidative damage at targets important for aging, like mitochondrial DNA, near the places of free radical generation. A low rate of free radical production can contribute to a low aging rate in animals that conform to the rate of living (metabolic) theory of aging. Available research indicates at least two main characteristics of longevous species: a high rate of DNA repair and a low rate of free radical production near mitochondrial DNA [48]. Further, overexpression with superoxide dismutase (SOD) and catalase has been obtained in 15 transgenic lines of Drosophila melanogaster flies, and the results indicated that eight of these showed an increase in MLSP (30% or less) [49]. 

The DNA of the nucleus in a human cell includes two strains of 3.1 × 10^9^ base pairs (bp) and consists of genomic information of 20,000 genomes, protein-coding genes. i.e., 20,000 human proteins are produced by the genomes. In addition to the DNA in the nucleus, the human mitochondrial DNA (mtDNA) was sequenced and revealed that the human mtDNA includes 16,569 bp and encodes 13 proteins, which are a part of the electron transport chain (ETC) located in the inner membrane of mitochondria [5,50]. Each cell has 100–1000 copies of mtDNA. It has been known that mitochondrial dysfunction causes so-called mitochondrial genetic diseases, such as, Alzheimer’s disease, CPEO (chronic progressive external ophthalmoplegia), diabetes mellitus, dystonia, KSS (Kearns–Sayre syndrome), Leigh’s syndrome, LHON (Leber’s hereditary optic neuropathy), MELAS (mitochondrial encephalomyopathy, lactic acidosis, and stroke-like episodes), MERRF (myoclonic epilepsy and ragged red fibers), mitochondrial myopathy, NARP (neurogenic muscle weakness, ataxia, and retinitis pigmentosa), Parkinson’s disease, and Pearson’s syndrome [51]. Recently, Progeria syndrome has also been believed to have caused mitochondrial dysfunction. The ETC is the complex that produces ATP by oxidative phosphorylation. During electron transport, 2–3% of electrons leak from the ETC, and superoxide is generated when they bind with molecular oxygen. In chain reactions, superoxide is converted to other reactive oxygen species (ROS) [5]. Among the ROS, peroxynitrite, made by the binding of superoxide and nitric oxide, should play a major role in generating oxidative stress within cells [5]. Peroxidation occurs at the same location, inside mitochondria. Rheumatoid arthritis, hepatitis, enteritis, cancer, aging, many neurologic diseases, and chronically persisting digestive diseases have recently been found to be related to oxidative stress. It is becoming evident that these are associated with the generation of ROS from the mitochondrial ETC (partially due to mitochondrial DNA, mtDNA, damage), subsequent lipid peroxidation, and cell death [5]. The mitochondrial genome has been shown to relate to aging as described [5]. In the different species, it is known that life span is different. Tolmasoff et al. measured superoxide dismutase-specific activity levels in cytoplasmic fractions of the liver, brain, and heart of two rodents (mus musculus (house mouse), Peromyscus maniculatus (deer mouse), and 12 primate species (Tupaia glis (common treeshrew), Saimiri scuireus (squirrel monkey), Galago crassicaudatus (bush baby), Saguinus mystak (mustached tamarin), Lemur macaco fulvus (lemur), Cercopithecus aethiops (African green monkey), Macaca mulatta (Rhesus monkey), Papio anubis (olive baboon), Gorilla gorilla (gorilla), Pan troglodytes (chimpanzee), Pongo pygmaeus (orangutan), Homo sapiens (man)). They found that a correlation exists between maximal life-span potential (MLP) and the ratio of superoxide dismutase (SODase) specific activity to specific metabolic rate (SMR) (SODase/SMR; SODase units per mg protein/SMR cal per g per day) in liver, brain, and heart tissues [52]. The SODase activity being measured was most likely from a mixture of the Cu/Zn and the Mn types of SODase enzymes found in the cytoplasm [53]. Ono and Okada further examined specific activities of six different enzymes, lactate dehydrogenase (LDH, EC 1.1.1.27), glucose-6-phosphate dehydrogenase (G6PDH, EC 1.1.1.49), SOD (EC 1.15.1.1), glutamic oxalacetic transaminase (GOT, EC 2.6.1.1), creatine phosphokinase (CPK, EC 2.7.3.2), and choline esterase (ChE, EC 3.1.1.7) in the cerebrums of 11 mammalian species—mouse, hamster, rat, guinea pig, rabbit, dog, sheep, swine, cat, bovine, and horse—to their maximum lifespans. The result showed that only the SOD level showed a positive correlation with maximum lifespans among the six enzymes they studied [54].

### 2.4. Aging Is a Phenomenon of Complexity

#### 2.4.1. Prolonged Evidence of Aging

The last common human ancestor to modern humans (H. sapiens) is representative of the earliest modern humans. It is suggested that modern humans arose between 260,000 and 350,000 years ago through merging populations in East and South Africa [55]. It is known that maternal inheritance is transferred through mitochondria [56], and paternal inheritance is transferred through the Y chromosome [57].

Life expectancy in Japan was 36.4 years in 1860, and over the next 160 years, it increased to 85.03 years in 2020. Although life expectancy has generally increased throughout Japan’s history, several drops were observed in the 1910s due to the Spanish Flu and the Second World War in the 1940s [58]

The Ministry of Health, Labour and Welfare (MHLW) released a summary of simplified life tables for 2021 on 29 July 2022. According to the table, in Japan in 2021, life expectancy was 81.47 years for men and 87.57 years for women. The 2013 data was the first time that life expectancy for males exceeded 80 years, and 2023 was the eleventh year that it has been above 80 years. In 2023, Japan ranked third oldest years old in the world (84.95 years old) after Hong Kong (85.83 years old) and Macao (85.51 years old) [59].

#### 2.4.2. Geroscience

The basic research on aging has been generalized as biogerontology, and started to be recognized as geroscience [60]. Kaeberlein et al. summarized ongoing projects for geroscience interventions with translational potential, i.e., dietary restriction, exercise, mTOR inhibitors, metformin and acarbose, NAD precursors and sirtuin activators, modifiers of senescence and telomere dysfunction, hormonal and circulation factors, and mitochondrial-targeted therapeutics. Interventions augment mitochondrial function, energetics, and biogenesis, including mitochondria-targeted antioxidants and the roles of NAD precursors [61]. This study focuses on mitochondria’s roles in aging and will be discussed.

### 2.5. Heteroplasmy-Self Classification and Self-Repair by Removal of Damaged Mitochondria

Aging is processed as an adjunct with cellular and physiological functional decline [62]. Accumulation of the damage through aging over time is thought to have a central role in the aging process. Recently, molecular damage in aging at the level of small molecules, proteins, RNA, DNA, organelles, and cells has been focused on [63]. Cell death control could be an important factor in regulating aging. Autophagy of cells may contribute to health, aging, and disease, and how to suppress age-associated diseases must be clarified [64].

Mitochondria may contribute to the aging process, but which components of mitochondria relate to the aging process must be considered. Gonzalez et al. hypothesize that humans with exceptional longevity harbor rare variants in nuclear-encoded mitochondrial genes (mito nuclear genes) that confer resistance against age-related mitochondrial dysfunction. They identify longevity-associated variants, genes, and mitochondrial pathways that are enriched with rare variants [65]. Lima et al. showed how mitochondrial stress pathways have pleiotropic effects on cellular and systemic homeostasis, which can comprise protective or detrimental responses during aging. They described defects in these conserved adaptive pathways, including regulating pathways of the mitochondrial unfolded protein response, mitochondrial membrane dynamics, and mitophagy. They emphasized that their failure contributes to heteroplasmy of mtDNA, and deregulation of key metabolites [66].

Recent studies show that the damaged DNA can be repaired [67]. The accurate maintenance of mtDNA is required for eukaryotic cells to assemble a functional electron transport chain. This independently maintained genome relies on nuclear-encoded proteins that are imported into the mitochondria to carry out replication and repair processes. Mitochondria employ robust and varied mtDNA repair and damage tolerance mechanisms in order to ensure the proper maintenance of the mitochondrial genome. The mtDNA repair and damage tolerance pathways includes base excision repair, mismatch repair, homologous recombination, non-homologous end joining, translesion synthesis, and mtDNA degradation in both yeast and mammalian systems [67].

### 2.6. Aging and Mitochondria

#### Mitochondria as Hallmark of Aging

While mitochondria are composed of about one thousand proteins, of which only 13 proteins are coded by mitochondrial DNA, those located in the electron transport chain (ETC), and proteins in mitochondria are encoded in nuclear genomes. In ETC, the numbers of mtDNA- and nDNA-encoded proteins are 7 and ~39 for Complex I (EC 7.1.1.2; NADH: ubiquinone oxidoreductase or Type I NADH dehydrogenase), 0 and 4 for Complex II (EC 1.3.5.1; Succinate dehydrogenase or succinate-coenzyme Q reductase, 1 and 10 for Complex III (EC 1.10.2.2; coenzyme Q: cytochrome c—oxidoreductase or cytochrome bc1 complex), 3 and 10 for Complex IV (EC 7.1.1.9; cytochrome c oxidase), and 4 and ~14 for Complex V (EC 7.1.2.2; ATP synthase) [68].

The biological basis of human aging remains one of the greatest unanswered scientific questions. The aging process results in various mtDNA mutations developed in individual cells, which causes age-associated mitochondrial dysfunction in an age-dependent manner, i.e., cells accumulate somatic mutations of mtDNA as part of normal aging and cancer development [69,70,71,72,73,74,75,76,77,78,79,80,81,82]. Jang et al. (2018) discuss three aspects of mitochondrial biology that link this ancient organelle to how and why human beings age: the role of mitochondria in regulating the innate immune system, the mechanisms linking mitochondrial quality control to age-dependent pathology, and the possibility that mitochondrial-to-nuclear signaling might regulate the rate of aging [77]. Jang et al. (2018) discussed three roles of mitochondria in aging. i.e., regulating the innate immune system, the mechanisms linking mitochondrial quality control to age-dependent pathology, and the possibility of mitochondrial-to-nuclear signaling [77]. They discussed the quality control mechanisms to deal with stress in the mitochondria. The magnitude of these mechanisms ranges in accordance, including the unfolding proteins repair of mitochondria (UPRmt) to initiate a transcriptional program to potentially relieve the stress, the removal of part of the mitochondria into a mitochondrial-derived vesicle (MDV) preserving the undamaged part, activation of mitophagy to remove the entire damaged mitochondria, and induction of cell death through apoptosis or necrosis to remove the entire damaged cell, titrating the magnitude of response as the level of perceived stress [77]. Mutation of mitochondrial DNA genome can damage cellular respiration, ultimately resulting in various progressive metabolic diseases collectively known as ‘mitochondrial diseases’. Mutagenesis of mtDNA and the persistence of mtDNA mutations in cells and tissues correlate well with the interplay of DNA replication, DNA damage and repair, purifying selection, organelle dynamics, mitophagy, and aging [75]. Thus, mitochondria have multiple functions and roles in aging.

### 2.7. Aging and Mitochondrial Damage

#### 2.7.1. Morphology of Mitochondria

Brandt et al. found that subpopulations of mitochondria from mouse liver show age-related changes in membrane morphology [76]. However, the same group found that mitochondrial function and ultrastructure are maintained in mouse hearts. They concluded that mitochondria are affected by aging depending on the organism [76].

#### 2.7.2. Mitochondria DNA Copy Numbers

Relatively limited attention has been devoted to measuring the number of mtDNA molecules per cell during aging. Using a PCR-based procedure, Miller et al. (2003) developed a precise assay that determines mtDNA levels relative to nuclear DNA [83]. Quantification was performed using a single recombinant plasmid standard containing a copy of each target DNA sequence (mitochondrial and nuclear). The copy number of mtDNA was determined by amplifying a short region of the cytochrome b gene (although other regions of mtDNA were demonstrably useful). Nuclear DNA content was determined by applying a single copy of the b-globin gene segment. The copy number of mtDNA per diploid nuclear genome in myocardium was 6970 ± 920, significantly higher than in skeletal muscle, 3650 ± 620 (*p* = 0.006). In both human skeletal muscle and myocardium, there was no significant change in mtDNA copy number with age from neonates to subjects older than 80 years [83]. Wachmuth et al. (2016) estimated the mtDNA copy number in 12 tissues of 152 individuals. They found age-related variation for two tissues: mtDNA copy number in skeletal muscle negatively correlates with age. In contrast, mtDNA copy number in the liver is positively correlated with age [84]. No corresponding results have been shown for mtDNA copy numbers through different ages. Over the last decade, evidence has suggested a causative link between mitochondrial dysfunction and major phenotypes associated with aging [85].

#### 2.7.3. Repair of Mutant mtDNA

In general, cells respond to DNA damage by DNA damage response pathways, which allow sufficient time for specified DNA repair to remove the damage physically. At least five major DNA repair pathways exist: base excision repair (BER), nucleotide excision repair (NER), mismatch repair (MMR), homologous recombination (HR), and non-homologous end joining (NHEJ) [86]. These repair processes are key to maintaining cell genetic stability. The DNA repair and damage-bypass mechanisms faithfully protect the DNA by either removing or tolerating the damage to ensure overall survival [86]. Fraga et al. showed that aging causes oxidative stress and increases 8-hydroxy-2′-deoxyguanosine (8-Oxisoguanine) in rat organ DNA and urine. They showed that oxidative damage to DNA during aging [87]. Additionally, de Souza-Pinto et al. (2001) examined the changes in DNA repair as measured by incision of 8-Oxoguanine in nuclear and mitochondrial DNA with aging, comparing 6 months (young mice) versus 14 months mice (old mice), and showed that 8-Oxoguanine glycosylase activity increases in mitochondrial but not in nuclear extracts in old mice [88]. de Souza-Pinto et al. (2009) showed that the incision increases in mtDNA but declines in nuclear DNA with increasing age. They identified the repair factor YB-1 as a key candidate for a mitochondrial mismatch-binding protein. This protein that localizes to the mitochondria of human cells contributes significantly to the mismatch-binding and mismatch-repair activity. When the intracellular levels of YB-1 are diminished, the mismatch-binding and mismatch-repair activity decreases, and when YB-1 depletes in cells, mitochondrial DNA mutagenesis increases [89]. Interestingly, it is known that the rate of mitochondrial mutagenesis is faster in mice than in humans [90]. These indicate that homeostasis, including mtDNA repair, is active in mitochondria rather than in nucleus. There is evidence for several DNA repair pathways in mitochondria, including BER, MMR, HR, and NHEJ [67,91,92,93,94]. In addition, the repair of mitochondrial DNA deletion is also described [95]. Bratic and Larsson (2013) described that increases in aging-associated mtDNA mutations are not caused by damage accumulation but rather are due to clonal expansion of mtDNA replication errors that occur during development [85]. Rong et al. (2021) described that mtDNA is more prone to be affected by DNA-damaging agents compared with nuclear DNA, and accumulated DNA damage may cause mitochondrial dysfunction and drive the pathogenesis of a variety of human diseases, including neurodegenerative disorders and cancer [96]. Fu et al. (2020) described that mitochondria respond to DNA damage and preserve their own genetic material in a manner distinct from that of the nucleus but that requires organized mito–nuclear communication. Failure to resolve mtDNA breaks leads to mitochondrial dysfunction and affects host cells and tissues [97].

#### 2.7.4. Mitochondrial DNA and Aging

Several experiments support the view that somatic mitochondria’s genetic and biochemical functions deteriorate during normal aging. Deletion mutations of the mitochondrial genome accumulate exponentially with age in the nerve and muscle tissue of humans and multiple other species [98]. A decline in electron transport activity with age and decreased bioenergetic capacity with age are shown [98]. Mitochondrial mutations may result from mitochondrial oxidative stress, and cells bearing pure populations of pathogenic mitochondrial mutations are sensitized to oxidant stress. Oxidant stress to mitochondria is known to induce the mitochondrial permeability transition, which has recently been implicated in the release of cytochrome c and the initiation of apoptosis. Thus, several lines of evidence support the contribution of mitochondrial dysfunction to the phenotypic changes associated with aging [98].

Lee and Wei (2001) described that defective mitochondrial turnover is a cause of accumulation of defective mitochondrial constituents and an important contributory factor to human aging [99]. Cytochrome c oxidase (COX) is the last enzyme in the respiratory electron transport chain of cells located in the inner membrane. Yu-Wai-Man et al. (2010) examined the distribution of COX-deficient extraocular muscle (EOM) fibers with age [100]. They found that COX deficiency increased depending on age in 46 patient samples. Wei (1998) examined 4977 mt mutation and lipid peroxidation in muscle, testis, and liver as a function of aging and showed this increase significantly with aging [101]. Tranah et al. (2018) measured leukocyte mtDNA m.3243A > G heteroplasmy(mtDNA mutation) in 789 elderly men and women from the bi-racial, population-based health, aging to identify associations with age-related functioning and mortality [102]. They found that the elevated heteroplasmy of mtDNA mutation was associated with reduced strength, cognitive, metabolic, and cardiovascular functioning. These indicate aging accelerates mitochondrial damage; mtDNA damage and deletion. Mitochondria appear and the quality change of mitochondria seems to limit the life span.

## 3. Conclusions

In this review, we focused on the role of mitochondria in the cause and development of aging in humans. Earth has 4.6 billion years of history, and one billion years after the Earth’s birth of Earth, living organisms were born. At that time, there was a small amount of oxygen in the air. Two billion years later, oxygen concentration increased, and the organisms had to adapt to the changing oxygen in the air. Organisms started to use oxygen to generate more intracellular energy, and multicellular organisms were born. Multicellular organisms suffer from aging in their lives. We focused on the mitochondrial damage during aging. In multicellular organisms. mitochondria produce intracellular energy, and mtDNA is concerned with life extent; the damage to mtDNA strongly relates to lifespan (Figure 5).

## Figures and Tables

**Figure 1 biomolecules-14-01317-f001:**
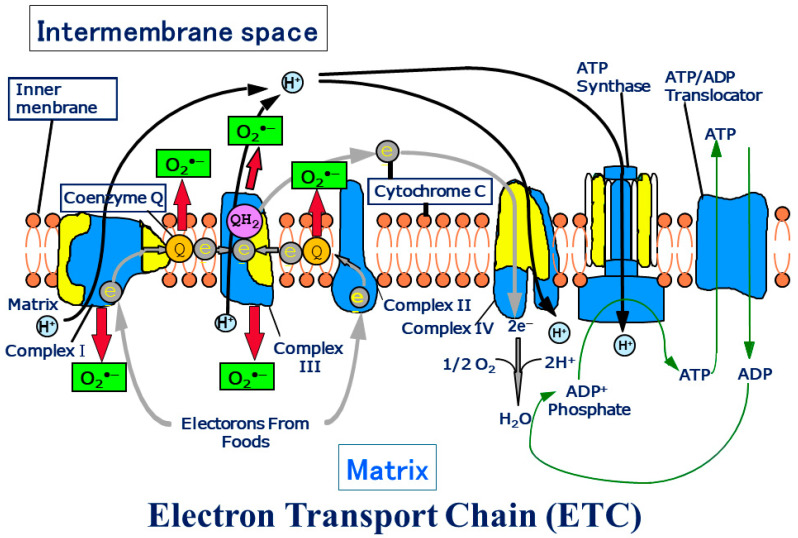
Schema of the electron transport chain. ATP production by oxidative phosphorylation with an electron transport chain (ETC).

**Figure 2 biomolecules-14-01317-f002:**
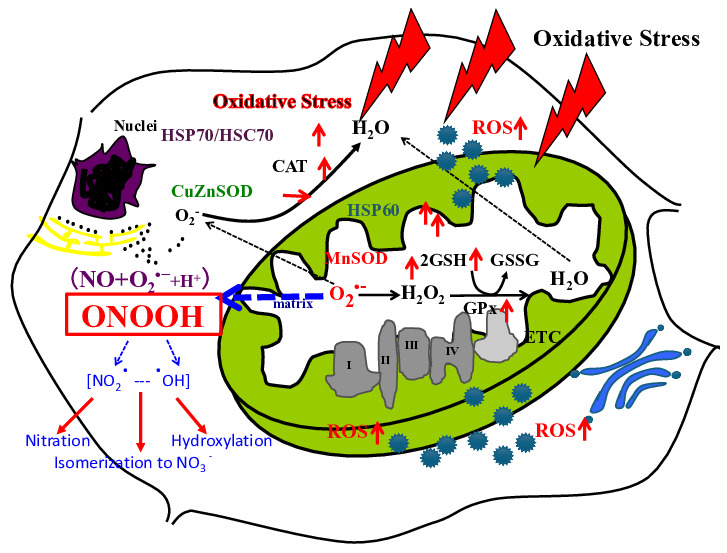
A scheme of mitochondrial generation of ROS. O_2_**^•^**^−^ generated from mitochondria binds with NO and further produces peroxynitrous acid (ONOOH), and further ONOOH produces NO_2_**^•^** and **^•^**OH. In the figure red arrow of up means up-regulations, and stable means no expression change. (Figure transferred from Reference [1]).

**Figure 3 biomolecules-14-01317-f003:**
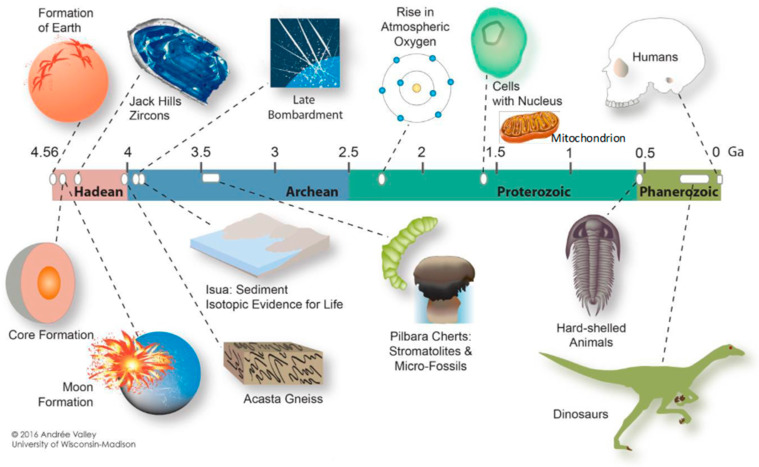
Illustration of the 4.6-billion-year-old Earth, acknowledging “Andrée Valley, University of Wisconsin—Madison”. (Reference [6]). The author, H.J.M. added mitochondria to the figure.

**Figure 4 biomolecules-14-01317-f004:**
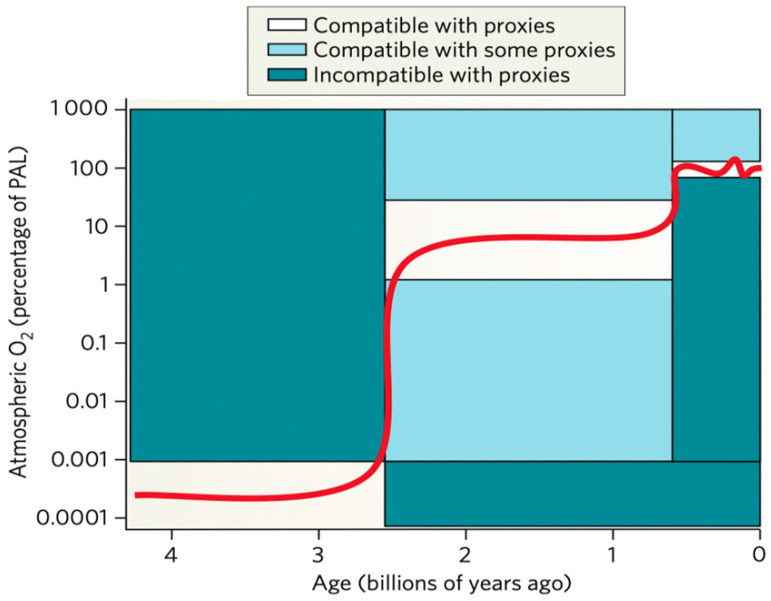
Increase of atmospheric oxygen in the history of Earth through 4.6 billion years. The figure was transferred from Reference [11] with a license.

**Figure 5 biomolecules-14-01317-f005:**
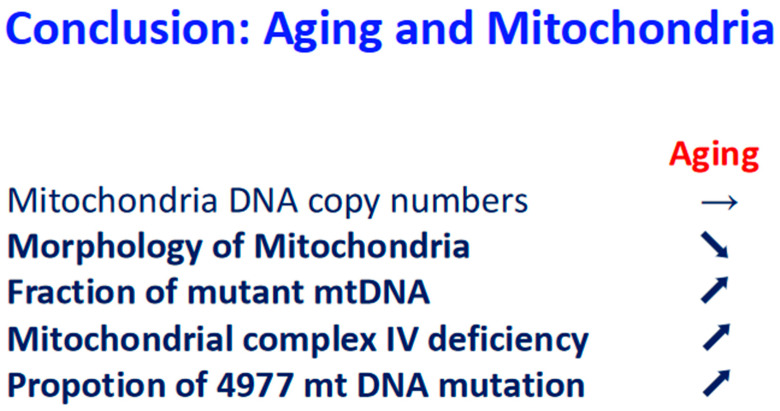
Conclusion: relationship between aging and mitochondria. Upon aging, arrow stable means no change, arrow down means down regulation, and arrow up means up regulation.

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
