# Peer review of "The Roles of Mitochondria in Human Being’s Life and Aging"

_biomolecules, 2024, doi:10.3390/biom14101317_

Round 1
Reviewer 1 Report
Comments and Suggestions for Authors
In this article, the author presents us with a fascinating and vivid story about the origin of mitochondria. After reading it, I even wonder if the author is a devoted fan of the National Geographic Channel. From the author's description, I have, for the first time, gained such a detailed understanding of how mitochondria emerged on ancient Earth. I must say, the author gave me a lively lesson in paleobiology.
Overall, I believe the article is divided into two parts. Sections 1-3 describe the relationship between changes in Earth's environment (mainly changes in atmospheric O2) and the origin of mitochondria, as well as the functional evolution of mitochondria. Sections 4-8 discuss the correlation between mitochondrial function and aging. Of course, these two parts are logically connected, which is not an issue. The issue, however, is that Biomolecular, in my opinion, is more focused on introducing the types and functions of biomolecules. I think sections 1-3 are more akin to popular science or promotional illustrations.
I am not proficient in paleobiology, geology, genetics, or geophysics, so I cannot judge whether the descriptions in sections 1-3 are objective, although the author does cite a lot of literature as evidence. Furthermore, regarding the origin of species and organelles, I believe there is ongoing debate in the academic community. Therefore, I feel that this section is more suitable to be presented as a promotional illustration, rather than as a review article describing scientific progress.
The writing in sections 4-8 is comprehensive, covering a great deal of specialized knowledge and experimental advancements. However, I would like to remind the author that some of the research findings are relatively outdated, and the references are somewhat old. It would be better to supplement with more recent research findings.
Additionally, in section 8.2, the author explains some changes that occur in mitochondria and mtDNA with aging. The author also elaborates on how mutations and variations in mitochondrial genetic material and proteins affect cells. However, I have a question: when it comes to mitochondrial biological behavior changes and aging, which is the cause and which is the effect? Although this may be a "chicken or egg" philosophical question, I still ask the author to discuss it appropriately.
I believe this is an excellent article, but I suggest the author consider splitting it into two parts: 1-3 and 4-8. I recommend keeping the second part in the current article and condensing and refining the first part to serve as an appropriate story background.
The author should review the entire text for some typographical errors. For instance, the abbreviation 'SQR' appears at lines 247 and 417, and similar errors occur with abbreviations like 'ROS' and 'SOD'. Also, there is a parenthesis after 'MLP' (line 340).
Author Response
Comment 1.: In this article, the author presents us with a fascinating and vivid story about the origin of mitochondria. After reading it, I even wonder if the author is a devoted fan of the National Geographic Channel. From the author's description, I have, for the first time, gained such a detailed understanding of how mitochondria emerged on ancient Earth. I must say, the author gave me a lively lesson in paleobiology.
A.1. Thank you very much for your comments.
Comment 2.: Overall, I believe the article is divided into two parts. Sections 1-3 describe the relationship between changes in Earth's environment (mainly changes in atmospheric O2) and the origin of mitochondria, as well as the functional evolution of mitochondria. Sections 4-8 discuss the correlation between mitochondrial function and aging. Of course, these two parts are logically connected, which is not an issue. The issue, however, is that Biomolecular, in my opinion, is more focused on introducing the types and functions of biomolecules. I think sections 1-3 are more akin to popular science or promotional illustrations.
A.2. As suggested, we created the new Section 1 (the old Sections 1-3), and Section 2 (the old Section 4-8), and 3. Conclusion.
Comment 3.: I am not proficient in paleobiology, geology, genetics, or geophysics, so I cannot judge whether the descriptions in sections 1-3 are objective, although the author does cite a lot of literature as evidence. Furthermore, regarding the origin of species and organelles, I believe there is ongoing debate in the academic community. Therefore, I feel that this section is more suitable to be presented as a promotional illustration, rather than as a review article describing scientific progress.
A.3.: Yes, there is ongoing debate in the academic community. Therefore, we added "There is ongoing debate in the academic community on the birth of living organisms on Earth." in the 1.3. Birth of living organisms on Earth section.
Comment 4.: The writing in sections 4-8 is comprehensive, covering a great deal of specialized knowledge and experimental advancements. However, I would like to remind the author that some of the research findings are relatively outdated, and the references are somewhat old. It would be better to supplement with more recent research findings.
A.4.: Thank you for your message "The writing in sections 4-8 is comprehensive". We rechecked references of 2. Mitochondria and human being' life.
Comment 5.: Additionally, in section 8.2, the author explains some changes that occur in mitochondria and mtDNA with aging. The author also elaborates on how mutations and variations in mitochondrial genetic material and proteins affect cells. However, I have a question: when it comes to mitochondrial biological behavior changes and aging, which is the cause and which is the effect? Although this may be a "chicken or egg" philosophical question, I still ask the author to discuss it appropriately.
A.5.: We added this sentence "Tranah et al. (2018) measured leukocyte mtDNA m.3243A > G heteroplasmy(mtDNA mutation) in 789 elderly men and women from the bi-racial, population-based health, aging to identify associations with age-related functioning and mortality [102]. They found the elevated heteroplasmy of mtDNA mutation was associated with reduced strength, cognitive, metabolic, and cardiovascular functioning." in 2.7.4. Mitochondrial DNA and Aging section. And we concluded, "These indicate aging accelerates mitochondrial damage; mtDNA damage and deletion." We added, "Since mitochondria appear, the quality change of mitochondria seems to limit the life span.".
Comment 6.: This is an excellent article, but I suggest the author consider splitting it into two parts: 1-3 and 4-8. I recommend keeping the second part in the current article and condensing and refining the first part to serve as an appropriate story background.
A.6.: Thank you. We followed your idea, split into three parts: 1. Oxidative stress, birth of Earth, and birth of living organisms on Earth, 2. Mitochondria and human being' life, and 3. Conclusion.
Comments on the Quality of English Language
The author should review the entire text for some typographical errors. For instance, the abbreviation 'SQR' appears at lines 247 and 417, and similar errors occur with abbreviations like 'ROS' and 'SOD'. Also, there is a parenthesis after 'MLP' (line 340).
A.7.: Thank you very much. We rechecked our manuscript and corrected it.
Reviewer 2 Report
Comments and Suggestions for Authors
Hiroko et al. Have produced an interesting overview entitled “The roles of mitochondria in human being’s life and ageing”. However, one can discuss if the title is appropriate. The manuscript covers several several topics, such as the mechanisms of mitochondria ROS production, birth of living organisms, the emergence of mitochondria 1.45 billion years ago, physiology of ancient mitochondria (1.6 billion years ago), lifespan in different species, and in the end of the manuscript about mitochondria and aging.
The article is interesting and easy to read. Parts of it is more like it was written for a popular science magazine, although the content is of good scientific quality. For those interested in history of the earth and evolution it is interesting, almost entertaining.
The weakness of the article is that it takes too long before we get to the promised part of mitochondria and aging. This part can be shortened. Some revision here is necessary. This reviewer would like to add another chapter on mitochondria and death.
Overall, I enjoyed reading the manuscript.
Author Response
Hiroko et al. Have produced an interesting overview entitled “The roles of mitochondria in human being’s life and ageing”. However, one can discuss if the title is appropriate. The manuscript covers several several topics, such as the mechanisms of mitochondria ROS production, birth of living organisms, the emergence of mitochondria 1.45 billion years ago, physiology of ancient mitochondria (1.6 billion years ago), lifespan in different species, and in the end of the manuscript about mitochondria and aging.
Comment 1.: The article is interesting and easy to read. Parts of it is more like it was written for a popular science magazine, although the content is of good scientific quality. For those interested in history of the earth and evolution it is interesting, almost entertaining.
A.1.: Thank you very much for your comments.
Comment 2.: The weakness of the article is that it takes too long before we get to the promised part of mitochondria and aging. This part can be shortened. Some revision here is necessary. This reviewer would like to add another chapter on mitochondria and death.
A.2.: Thank you. We deleted some parts "Reduced melts having oxygen close to that defined by the iron-wüstite buffer would yield volatile species such as CH4, H2, H2S, NH3, and CO, whereas melts close to the fayalite-magnetite-quartz buffer would be similar to present-day conditions and would be dominated by H2O, CO2, SO2 and N2. Direct constraints on the oxidation state of terrestrial magmas before 3,850 Myr before the present (that is, the Hadean eon) are tenuous because the rock record is sparse or absent.".
Comment 3.: Overall, I enjoyed reading the manuscript.
A.3.: Thank you very much for your comment.
Round 2
Reviewer 1 Report
Comments and Suggestions for Authors
The author addresses most of my concerns. Therefore, I believe that in its current state, although it is not yet perfect, the manuscript is worth recommending for researchers to read.
Comments on the Quality of English LanguageUnderstandable.